# Hardware-Intrinsic Multi-Layer Security: A New Frontier for 5G Enabled IIoT

**DOI:** 10.3390/s20071963

**Published:** 2020-03-31

**Authors:** Hussain Al-Aqrabi, Anju P. Johnson, Richard Hill, Phil Lane, Tariq Alsboui

**Affiliations:** 1Department of Computer Science, Centre for Industrial Analytics (CIndA), School of Computing and Engineering, University of Huddersfield, Queensgate, Huddersfield HD1 3DH, UK; r.hill@hud.ac.uk (R.H.); p.lane@hud.ac.uk (P.L.); T.Alsboui@hud.ac.uk (T.A.); 2Department of Engineering and Technology, Centre for Planning, Autonomy and Representation of Knowledge (PARK), School of Computing and Engineering, University of Huddersfield, Queensgate, Huddersfield HD1 3DH, UK; a.johnson@hud.ac.uk

**Keywords:** Internet of Things, cloud computing, hardware security, field programmable gate array (FPGA), 5G, analytics

## Abstract

The introduction of 5G communication capabilities presents additional challenges for the development of products and services that can fully exploit the opportunities offered by high bandwidth, low latency networking. This is particularly relevant to an emerging interest in the Industrial Internet of Things (IIoT), which is a foundation stone of recent technological revolutions such as Digital Manufacturing. A crucial aspect of this is to securely authenticate complex transactions between IIoT devices, whilst marshalling adversarial requests for system authorisation, without the need for a centralised authentication mechanism which cannot scale to the size needed. In this article we combine Physically Unclonable Function (PUF) hardware (using Field Programmable Gate Arrays—FPGAs), together with a multi-layer approach to cloud computing from the National Institute of Standards and Technology (NIST). Through this, we demonstrate an approach to facilitate the development of improved multi-layer authentication mechanisms. We extend prior work to utilise hardware security primitives for adversarial trojan detection, which is inspired by a biological approach to parameter analysis. This approach is an effective demonstration of attack prevention, both from internal and external adversaries. The security is further hardened through observation of the device parameters of connected IIoT equipment. We demonstrate that the proposed architecture can service a significantly high load of device authentication requests using a multi-layer architecture in an arbitrarily acceptable time of less than 1 second.

## 1. Introduction

Adopting evolving business models that are enabled by emerging 5G technologies is a challenge when attempting to maintain legitimate security and privacy considerations for Internet of Things (IoT) and Industrial Internet of Things (IIoT) devices  [1]. It is clear that raising industrial users’ knowledge that a substantial amount of the interest they create is intrinsically connected with intellectual property (IP) ownership and continuous development. There is also the persistent risk of a security breach that could compromise ownership of the IP, putting the underlying business model at higher risk [2,3]. This is particularly prevalent in the provision of IoT assisted healthcare systems, which is a pertinent example of distributed IT systems that have similarly complex needs and stakeholder requirements  [4]. Although cloud computing illustrates how technologies and business models are used to provide new business opportunities to enterprises, businesses remain at risk of emerging threats due to the proliferation of cloud services, including multi-tenant cloud environments [5,6]. The promise of 5G Infrastructure holds immense possibilities for greater integration of physical devices that are ideally suited to IIoT for several reasons, as follows:*Less network latency* increases overall response times and is able to enhance security protocol strictness without sacrificing the system’s user experience;*Higher data rates* enable the sharing of data between devices, and the utilisation of metadata to support secure transactions building trust between devices;*Lower power demand* allows widespread use of sensing and processing devices where power infrastructure is absent.

The huge advantage of millimetre wave (MMW) radio spectrum for 5G is a crucial enabler for better network performance, although at a loss of propagation range [2]. Whereas the higher frequency band has specific physical security [7,8], this approach is not one that we should depending on. A manipulative attacker seated beside the IIoT device may be able to transmit data externally  [9,10,11,12]. The heterogeneous nature of IoT communications with its heterogeneous architecture and devices, requires information sharing and collaboration across a wide range of networks. This poses severe privacy and security issues [13]. IoT privacy protection seems to be more vulnerable than conventional Information and Communication Technology (ICT) systems because of several vector threats against IIoT technologies [14,15]. Modelling these vulnerabilities is challenging, particularly since the multiplicity of IIoT devices each represent agents within a complex system of interactions that need to be secure [16,17,18].

Consequently, there is a need to create a flexible multi-layer cloud security architecture that provides adequate authentication for multiple parties in a reliable way, while being mindful of the heterogeneous nature of how IIoT devices will communicate efficiently. Our article discusses how well the cloud methodology was developed to guide the creation of the security architecture for several purposes. Firstly, cloud computing architectures actively support complex demands via elasticity  [19,20] and facilitates the standardisation of diverse systems by abstraction. Secondly, there seems to be a proven architectural reference model given by NIST [21], which is widely used. Lastly, cloud systems have similar features with IIoT systems in that multiple parties need to function together and collaborate by a secure exchange of data and assets [5].

Previous work addressed the specific instance of multi-party trust authentication for the deployment of cloud based business intelligence systems. The authors also have built and adapted to accommodate a particular instance where the introduction of 5G network services would enable new business opportunities through increased efficiency. To support these features, the authors extended cloud-based infrastructure to include Physically Unclonable Function (PUF) hardware. Since the PUFs are resilient to spoofing attacks, the PUF hardware offers a higher level of security toward direct physical attacks, which are essential in situations where there is a need to rapidly authenticate several parties to ensure trustworthy connections [5].

The delivery of analytical resources from manufacturing plant represents a real scenario that the authors addressed, allowing the secure exchange of heterogeneous data, and also performance appraisal, between both the IIoT components and the enterprise (ICT) system of the organisation, often using Micro Services architecture [22]. This article considers the potential adversarial attacks to consider on such a device, which assists the design of an agile approach to multi-layer security. The authors created algorithms that require authentication through PUFs to provide effective, secure, and flexible access to IoT cloud applications. The article is arranged as follows. Section 2 defines a framework for multi-layer security. Section 3 presents a related secure solution for networking which utilises PUFs. In Section 4, we present the results of experiments that illustrate the potential for this approach. Finally, we conclude in Section 5.

## 2. Multi-Layer Security Model

The critical challenge for IIoT is the implementation and processing the large amount of data produced by these devices. In attaining this IoT vision, Low-Power (LP) and Loss Networks (LLNs) are diversified, and the interconnection of restricted physical devices by the use of LP and LLNs involves the modification of protocols and existing structures currently in common use [23]. Latterly, hardware trojan attacks have developed as a threat to all hardware and integrated circuits (ICs) [24]. The main challenge of handling network connectivity in a tightly-equipped setting, including a smart factory, is to identify and manipulate different attack vectors. In principle, the promise of cloud resources also introduces potential system vulnerabilities. As such, the authors opted to create a security model that divides a variety of security controls through multiple layers of defences [25]. Figure 1 illustrates a proposed secure architecture. The authors use the example of a traditional enterprise infrastructure with analytics capabilities to promote tactical and organisational business decision-making.

Primarily, as just that, our model was examined, in which individual users are tenants in a multi-tenant cloud environment. In our model, we consider the case that each user or (IIoT device or sensor) is described by a multi-cloud enterprise system as a prospective tenant. As the architecture enables the abstraction of resources, users that also require access to the business network can do this remotely, through virtual machines, and also through hardware devices [25].

All endpoints are secured via firewalls. In the beginning, all external requests are assisted by authentication data firewalls for each potential tenant. The Metadata layer, for example offers security controls for the features previously allowed for each tenant registered. The lack of required authentication data will prevent the user from effectively communicating with the system. Once the simple authentication is established, a Tenant Metadata Layer maintaining rules-based controls is required to determine which part of the business system a permitted tenant can access. For instance, this may apply to specific databases or reports. While the IIoT device offers data for a variety of analytics processing, which involves not only adding data to the repository as well as maintaining access to other data sources which can be collected and merged to present better analytical services.

A secure connection must be established, and this has been achieved by using the public-key Infrastructure (PKI). The PKI uses it to verify that the signature is authentic. Within its model layer, public key certificates are preserved within the Digital Vault, and this offers another secure degree where the user session may be approved or removed. In the case where the deceptive attackers penetrated aggressively into the first three layers, Layer four offers a deeper layer of protection. Whereas the controls of the prior layers are capable of protecting against various attacks, these can not prevent them from a harmful intruder who previously has the authority to access the system. The network will monitor suspicious activities using the Intrusion Prevention System as well as to detect irregular actions, in order to set up a session for tenants engaging in inappropriate behaviour.

An anti-malware layer of protection reinforces layer four. Far more surreptitious activity, for example, hidden executable code, may disrupt as it is implemented into the business network. Layer 5 keeps an activity record, and a list of known threats. Within the application cloud layer, this layer comprises the business features and is of considerable value to enterprise clients. During that time, the client has entered this layer, simple authentication, client verification by PKI, intrusion prevention system (IPS) and anti-malware inspections were already made, with each layer being able to terminate the session. Apart from business applications, it is necessary to access corporate repositories by a particular type of user, whether directly through application programming interfaces (API) or through querying and monitoring interfaces, usually provided via a web portal [26].

## 3. NIST Cloud Model

NIST is developing standard protocols and guidelines for user or client devices access to the Cloud by means of an interface for virtualisation, Internet browser interface, and the thin client interface [27]. These clouds are formed of a 7-layer architecture, consisting of layers: (1) as the layer of the physical infrastructure components, (2) layer of resources abstraction for virtualisation, (3) the layout layer for virtual Services, (4) the infrastructure as a service (IaaS) layer, (5) the layer for platform as a service (PaaS), (6) the application layer of software as a service (SaaS), and (7) the layer of applications for the tenants. The proposed multilayer security model may be compared to the cloud model of NIST [27] as follows. In NIST layers there are tenant users which could be hardware devices or virtual machines (VMs). Such a model may be implemented to each layer according to the principles of trustworthy computing [1,2].

Each session is aligned to layer six through a sequence of authentication and verification phases in the fourth and fifth layers. For applications which are hosted off-premises, layer seven access is made available via API interfaces. The presence of a firewall suggests infrastructure as a service (IaaS) [20], whereas management systems exist within the platform as a service (PaaS) layer. Software applications will reside in a software as a service (SaaS) tier.

### Session Workflow

A typical session workflow is illustrated in Figure 1. The allocation of session IDs in layers three and two contributes to the setup of a new client by a future IIoT tenant user. This would be accompanied by the access identifier given in layer four. Following this stage, where the inspection of packets is a crucial task for each of the sessions that have taken place so far. The database of metadata (DB_META_) and database of vault (DB_VAULT_) layers require the verification of IIoT requests before the packet inspection is performed for each session using a database of intrusion prevention systems DB_IPS_ and database of anti-malware DB_ANTIMAL_. The DB_IPS_ and DB_META_ link explicitly to PaaS functions within the context of the NIST model. In comparison, in the model Database of firewall DB_FW_ is known as IaaS. Supplementary authentication is required for each SaaS user, although at this stage, there is still a substantial number of verifications. Nevertheless, this verification is intended to enforce the company structure role-based permissions, such as the sub-set of employees, which offer access to the sensitive payroll information, for organisational data protection.

## 4. Hardware-Intrinsic Secure Multi-Layer Connectivity Model

The presented model considers users requesting access to services such as analytics in industrial infrastructure. Due to advancements in hardware technologies, users, as well as IIoT application services, incorporate hardware platforms on a large scale. One such advancement is the use of FPGAs solutions with hardware and software programmability providing flexibility and scalability to address IIoT requirements [28]. Applications such as data processing are an unavoidable part of IIoT, and FPGAs are an invaluable part of meeting future processing demands. FPGA-based data centres provide a volume of computation and storage resources to be efficiently processed on the edge of the network.

The FPGA based accelerations have significant potential for industrial applications enabling real-time data processing by combining locally generated data with additional enterprise data. Hardware acceleration, flexibility, and performance provided by FPGAs are an attractive solution for 5G networks for meeting the changing and increasing demands of the wireless markets. Currently, FPGAs provide optimised solutions for 5G technologies such as cloud-based radio access network (cRAN ), virtual radio access network (vRAN), Massive multiple-input, and multiple-output (MIMO), Backhaul, Fronthaul, Digital Radio Front-End  [29].

With the rising number and connectivity of IIoT intelligent devices with the network, the model requires to process an increased volume of transactions. To deal with an increased processing volume, the multi-layer model requires compliance, where the proposed system dynamically provides the required flexibility and security. Below we describe the procedure adopted to introduce an IIoT device with an inbuilt design feature, which increases the level of security with the connecting components. We use the concept of hardware-intrinsic security, which develops security from the intrinsic properties of the silicon. The security primitive employed in this work is Physically-Unclonable Functions (PUF), which utilises intrinsic manufacturing differences in the electronic hardware for strengthening security.

We describe a protocol for secure connectivity in the network. The protocol introduces a series of steps that permit all new clients entering the IIoT system. To grant access to the IIoT system, a current client needs to introduce the new customer following a series of procedures (Algorithm-2), as described below. The model consists of *K* verification layers. Verification at each layer is assured using a PUF based security protocol. Every layer of the security model has a PUF. In the multilayered model K=7, there exist a PUF for each existing user in each layer, which represents the fingerprint of every existing genuine member of the IIoT node. In this work, we use FPGAs for implementing the PUF. Packaged as a cloud manager, it generates a composite PUF and model ($M_{A}$), that represents the physical PUFs in the *K* layers of the model. Genuine clients receive an obfuscated bitstream consisting of a description of the mathematical PUF model through a secure communication channel. The genuine user introduced by an existing customer then downloads the bitstream and implements the PUF. We describe a PUF Based Authentication Protocol for verifying a client.

The cloud management plane handles the authorisation request initiated by the client $U_{A}$. The authorisation is processed by a security check involving the PUF, where *q* challenge sets($CH_{p}$) of length *n* is sent to $U_{A}$ together with a random number (rand). The received challenge bits are presented to the PUF model ($M_{A}$) at the client’s end, and the corresponding responses are collected for every layer in the proposed model.As we are considering a *K* layer model, there reside *K* responses which represent a single challenge string for every layer. A pre-agreed shuffling scheme is used to scramble the entire responses (K.q) for all challenge sets. The client and the management plane accord with an encoding E(.) and decoding D(.) scheme sor secure transmission of PUF responses. The user $U_{A}$ then sends the scuffled responses encoded with E(.) to the cloud model for confirmation, and the clould management layer decodes with D(.) to convert the response back and direct the responses to the respective cloud layer.

The original challenge bits of *q* are then added to the actual PUFs existing in the layers of the IoT cloud, and the responses are collected. The mathematical PUF and physical PUF responses are examined for a high similarity to declare the user $U_{A}$ to be genuine.

The quality of the PUF is mainly determined by two parameters, which are reliability and security. A reliable PUF has a sufficiently long lifetime and provides a stable response under different external circumstances. Considering minuscule variations in XOR PUF responses, the presented algorithm provides a tolerance of 1% in PUF response comparison. The parameter security addresses the level of protection that a PUF offers against a wide range of attacks. We ensure security by employing a powerful Arbiter PUF with >10 component XOR-PUFs stages to enhance security and to counter machine learning interventions  [30].

### 4.1. Algorithm Design

To strengthen security, PUF based verification supplements the existing verification in the primary cloud multi-layer model. FPGAs residing in cloud layers contain PUFs describing all existing clients. Additionally, an existing genuine client comprises a mathematical model of PUF which is transferred from the cloud management unit. The mathematical model is implemented in the client FPGA following Dynamic Partial Reconfiguration. The mathematical model is constructed by the IIoT infrastructure using machine learning as it has access to internal parameters of constituent Arbitor PUF stages. To guaranty security, a strong PUF is employed in the system. A strong PUF promises to provide resistance to cloning by a malicious adversary adopting machine learning approaches [31]. This security is ensured by increasing the constituent Arbitor PUF stages to greater than 10.

Algorithm 1 describes the process to be followed to grant a request to access to an application. Various checks followed in the cloud model is represented in each step of the algorithm. The proposed model is flexible to incorporate additional security if required by provision to extend the security to additional layers. A database of previously used challenges is maintained by the cloud management unit to prevent repeated usage of same challenge bits and ensure security gained replay attacks.

The model contains physical PUF representing the client in each layer of the cloud model and mathematical PUF, describing the functionality of the physical PUF. An obfuscated bitstream is used to download the mathematical model of physical PUF using DPR. The mathematical model is constructed by the IIoT infrastructure using machine learning as it has access to internal parameters of constituent Arbitor PUF stages.

Robustness is provided by a strong PUF, which cannot be cloned by malicious third parties. The requirements for the PUF considered in this work include (A) a Strong PUFs with a vast number of possible challenges, (B) unpredictability of challenge responses which means the difficulty to extrapolate or predict the CRPS from the known CRPs.

The security is ensured by increasing the constituent Arbitor PUF stages to greater than 10. Algorithm 2 provides authentication for all user requests for entry to the IIoT system. Each step of the process delivers security checks required at each stage of the layer. The algorithm is briefly described below. A set of challenges are generated, which excludes prior sets, and these are used between client and cloud layers during their authentication interactions. Each authentication requires a collection of *q* challenge bits, each being length *n*. Both the mathematical model and the physical model are provided with the same to generate the responses. At each cloud layer, the produced responses of the mathematical PUF and the physical PUF are compared for verification. The high similarity of responses (>=99%) is considered genuine, and the client is granted to proceed to the next layer of security checks. A database of previously used challenges bits is maintained to disregard any repeated usage, which would otherwise provide a chance for a replay attack. For a challenge set size of *q* and length *n*-bit used for each authentication attempt, provides ($2^{n}/q$) possible attempts of access on the application. The challenge bit-size is extensively large, requiring billions of years to be completely exhausted.
**Algorithm 1** Multi-layered security model using PUF: New client**Objective:**(a)The seven layer cloud model consisting of FPGA clouds verifies the identity of a new client FPGA ($U_{B}$) who is requesting access.(b)The cloud model provides application access for the genuine client ($U_{B}$).**Prerequisites:**(a)New client $ClientU_{B}$, requesting application access is known to an existing client $U_{A}$ as a genuine applicant.(b)Cloud-FPGAs have built-in controllers to facilitate secure dynamic partial reconfiguration.(c)Client-FPGA has built-in controllers to facilitate secure dynamic partial reconfiguration initiated by the cloud.(d)The Cloud-FPGA fabric is divided into two parts, (a) static fabric and (b) dynamic fabric. Static fabric consists of hardware configurations which existed before deployment. The dynamic fabric of the Cloud-FPGA is dedicated to configure additional security primitives (mostly PUFs) for any genuine clients using secure dynamic partial reconfiguration.(e)The client-FPGA fabric is divided into two parts, (a) static fabric and (b) dynamic fabric. Static fabric consists of hardware configurations which existed before deployment. The Client-FPGA has secure remote DPR controllers in the static partition facilitating configuration of PUF mathematical model in the dynamic fabric, via an obfuscated bitstream.**Input:**$P_{CT}$, $DB_{FW}$, $DB_{META}$, $DB_{VAULT}$, $DB_{IPS}$, $DB_{ANTIMAL}$ of $UserU_{A}$(a)Tenant session: *S*(b)Contents of session packets:$P_{CT}$(c)Contents of FW: $DB_{FW}$(d)Contents of $TENANT_{META}:DB_{META}$(e)Contents of $TENANT_{VAULT}:DB_{VAULT}$(f)Contents of $IPS:DB_{IPS}$(g)Contents of $ANTIMALWARE:DB_{ANTIMAL}$Note: $DB_{j}$ represents content DB of layer *j***Output:** A value in Flag to show a successful dynamic partial reconfiguration (Flag=1) or denied (Flag=0).**Steps:**1.Initialize S=1, E=12.$U_{i}$ to management plane MP: request access to application *A*3.MP to $U_{i}$: MP sends a random number rand and a set of challenges $CH_{p}$ consisting of *q* challenge bits each of length ‘*n*’.4.$U_{i}$ calculates the following:
$Rim_{p,j}=Mi\left(CH_{p,j}\right)$, p=1…q, j=1…K$Rim=\{Rim_{p,j}$, 1≤p≤q, 1≤j≤K}$CA_{i}$ = $S\left(E(Rim),rand\right)$
5.$U_{i}$ to MP: certificate $CA_{i}$6.**foreach** layer *j***do**(a)Initialize Mem=0, Match=0(b)**If** (E=1)
(a)$MP:Rim_{p,j}=S^{\prime }\left(D(CA_{i}),rand\right)$(b)MP to $Cloud-C_{i}$: Set of challenges $CH_{p}$ and $Rim_{p,j}$(c)$Cloud-C_{j}$ calculates the following
$Rif_{p,j}=Pi\left(CH_{p,j}\right),p=1\ldots q,j=1\ldots K$**if**Nij≥0.99Mem=1
(d)**if** ($P_{CT}$$\in DB_{j}$, |$DB_{j}\in \{DB_{FW},$$DB_{META},$$DB_{VAULT}\}$ AND $P_{CT}\notin DB_{j}$, |$DB_{j}\in \{DB_{IPS},$$DB_{ANTIMAL}$} ); Match=1i.**if** (Mem&&Match), E=1; proceed to next higher layerii.**else** Exit; set E=0, S=0; DenyTenantAccess()7.**if**S=1; AuthoriseTenantAccess()
**Algorithm 2** Multi-layered security model using PUF: Client is an existing User [2]**Objective:**The seven layer cloud model consisting of FPGA clouds verifies the identity of a client FPGA ($U_{A}$) who is requesting access.The cloud model provides application access for the genuine client ($U_{i}$).**Prerequisites:**An *n*-bit input, 1-bit output XOR PUF $P_{1}$ is reconfigured in all layers of the Cloud-FPGA. There exists a PUF for every authenticated user. PUF $P_{ij}$ represents the identity of the user *i* in the cloud layer *j*.A combined mathematical model $M_{i}$ representing all the *K* PUFs in the cloud layers, resides with each user $U_{i}$.Cloud-FPGA and user $U_{i}$ have agreed on a fixed encoding scheme E() and a decoding scheme D(.), such that for any binary string x,E(.) and D(.) are injective, X=E(x) and D(X)=x.Cloud-FPGA and user $U_{i}$ have agreed on a shuffling scheme Y=S(X,rand), and $S^{\prime }\left(Y,rand\right)=X$ where rand is a random number.**Input:***S*, $P_{CT}$, $DB_{FW}$, $DB_{META}$, $DB_{VAULT}$, $DB_{IPS}$, $DB_{ANTIMAL}$Tenant session: *S*Contents of session packets:$P_{CT}$Contents of FW: $DB_{FW}$Contents of $TENANT_{META}:DB_{META}$Contents of $TENANT_{VAULT}:DB_{VAULT}$Contents of $IPS:DB_{IPS}$Contents of $ANTIMALWARE:DB_{ANTIMAL}$Note: $DB_{j}$ represents content DB of layer *j***Output:**A value in variable *S* to show that the application access is granted (S=1) or denied (S=0).**Steps:**1.Initialize V=1, E=1, Flag=02.$U_{B}$ requests $U_{A}$, for an introduction to access application *A*3.$U_{A}$ to MP: request introduction of $U_{B}$ to cloud layers $C_{j}$4.MP to $U_{A}$: MP sends a random number rand and a set of challenges $CH_{p}$ consisting of *q* challenge bits each of length ‘*n*’.5.$U_{A}$ calculates the following:
$RAm_{p,j}=MA\left(CH_{p,j}\right)$, p=1…q, j=1…K$RAm=\{RAm_{p,j}$, 1≤p≤q, 1≤j≤K$CA_{A}$=$S\left(E(RAm),rand\right)$
6.$U_{A}$ to MP: certificate $CA_{A}$7.**foreach** layer *j***do**(a)Initialize Mem=0, Match=0(b)**If** (E=1)
i.$MP:RAm_{p,j}=S^{\prime }\left(D(CA_{A}),rand\right)$ii.MP to $Cloud-C_{j}$: Set of challenges $CH_{p}$ and $RAm_{p,j}$iii.$Cloud-C_{i}$ calculates the following
$RAf_{p,j}=PA\left(CH_{p,j}\right),p=1\ldots q,j=1\ldots K$$N_{Aj}=\left(1-\frac{\sum_{\left(p=1\right)}^{q}(R_{Amp}⨁R_{Afp}}{q}\right)$**if**NAj≥0.99Mem=1
iv.**if** ($P_{CT}$$\in DB_{j}$, |$DB_{j}\in \{DB_{FW},$$DB_{META},$$DB_{VAULT}\}$ AND $P_{CT}\notin DB_{j}$, |$DB_{j}\in \{DB_{IPS},$$DB_{ANTIMAL}$} ); Match=1v.**if** (Mem&&Match), E=1; proceed to next higher layervi.**else** Exit; set E=0, Flag=08.**if**V=1; Verified introducing client
(a)**foreach** layer *j***do**iCloud-FPGA, $C_{j}$ initiates DPR and configures a new PUF $P_{B,j}$, PUF $P_{B,j}$ represents the identity of the $U_{B}$ in the cloud layer *j*ii$C_{j}$ to MP PUF modeling parameters $param_{j}$(b)MP generates a combined Mathematical model $M_{B}$ of all PUFs $P_{B,j}$ in the cloud layers(c)MP generates obfuscated bitstreams of PUF mathematical model $M_{B}$(d)MP initiates remote dynamic partial reconfiguration of PUF $M_{B}$ in the dynamic partition of the client-FPGA$U_{B}$(e)Flag=1 and exit; follow protocol-1. $U_{B}$ is same as any other existing client.

Additionally, our model is fully flexible with DPR capability, which permits new security primitives, including novel PUF architectures, to be replaced with the existing once. This further increases the life span of the model. In addition, considering the research in the area [7], the possibility of repeated challenges occurring is highly unlikely for comparable challenge set volumes.

The presence of new users being proposed by an pre-existing authenticated system users is made possible by the sharing of model responses. Once the existing client has been successfully authenticated, new PUFs are created by the FPGA at run-time. After DPR, the mathematical PUF model is downloaded to the new user FPGA using an obfuscated bitstream. Again, the security model makes the assumption that the security process of maintaining system integrity is followed by a secure DPR process. The new cloud user will then use an algorithm of 1 to obtain the application layer.

### 4.2. Device Parameter Analysis of Client FPGAs

A genuine client, which turns to be potentially malicious by modifying the client device architecture to attack the IIoT application, is confirmed by client device parameter verification using neural networks. The client analysis process strengthens the security of the IIoT application against potentially malicious clients. A legitimate cloud service requires each IIoT client FPGA to satisfy specific requirements. Firstly, the FPGAs require a Dynamic Partial Reconfiguration (DPR) ability, which facilitates the set up of PUF primitives in the FPGA fabric. DPR allows dynamic reconfiguration of hardware units in selected regions on the FPGA framework.

The FPGA floor plan requires dynamic partitions that promote analysis of the fabric by the cloud service. Although DPR offers tremendous flexibility for IIoT applications, security needs to be ensured to avoid DPR based Trojan insertions as proven in [7].

An additional security measure adopted in the proposed scheme is to analyse the device parameters of the client FPGA. DPR performs this by sending an obfuscated and downloadable bitstream which collects the client device parameters. The device parameters are tested at the malware detection layer to assure that variations in the attributes of the FPGA clients. A client FPGA signature is a mechanism for identifying malicious adversaries. An initial DPR process implements a design that collects the device parameters, and a second DPR process erases the downloaded bitstreams. These device parameters are directly collected by the cloud management unit to evade manipulations from the client.

The proposed architecture for device parameter verification for Trojan analysis is shown in Figure 2. The design consists of two layers of neurons, where the input layer neurons (LAYER-1), produces a spike train with a frequency proportional to the device parameter. The number of input layer neurons represents to the number of device attributes that are analysed. The second layer neuron responds based on the spike rate received from its presynaptic neurons. In Figure 2, *K* input layer neurons are shown. There exist eight parallel connections between two tiers of pre-post synaptic neurons. This is to mimic the parallel connection between neurons in the brain-inspired systems, which aids in building post-synaptic potential and enhances fault-tolerance. The pattern identification procedure regulates the spike rate between layer 1 and 2. This is depicted using the Gaussian distribution shown between the neurons.

The distribution represents a variable transmission probability depending on the particular pattern. The nomenclature $PR_{Kr}$ represents transmission probability between presynaptic neuron *K* and the post-synaptic neuron in the $r^{th}$ interconnection between the pair. The output layer neuron provides a stable enable signal for the client FPGA if the received device parameters are within scope. This principle of using a spiking neural network is derived from [32,33], and hardware realization of the approach is described in [34].

However, in [32,33,34], the authors derive bio-inspired principles for homeostasis targeting robotic applications, where this paper emphasis the use of similar methodologies for hardware Trojan detection. Bio-inspired computing develops computational models using various models of biology. Brain-inspired computing is a subset of bio-inspired computing, which is mainly based on the mechanism of the brain. Brain-inspired models help to narrow the hardware Trojan detection process based on the mechanism of the brain, which produces a compact computational model rather than the complex biological process involved in the former. A pattern identification protocol verifies the pattern where spike to the postsynaptic neuron (LAYER-2) is regulated using a transmission regulation following a Gaussian relation. A high transmission probability (PR) is provided by the transmission regulation unit provided the device parameters are in the acceptable range.

A lower PR indicates a more significant deviation from the device parameter standards and fewer input spikes arriving at LAYER-2, which provides a stable firing rate by following Spike-timing-dependent plasticity (STDP) [35] and Bienenstock–Cooper–Munro (BCM) learning rules [36] for spike rates in the permissible range. Otherwise, the postsynaptic spikes drop to zero. Multiple connections are laid between each pre-post synaptic neurons to increase the security of the detection unit from any intruder from attacking the Trojan analysis unit.

## 5. Experimental Results

In this work, we implemented an XOR PUF construct consisting of 10 parallel Arbiter PUFs with 64 switch blocks on Xilinx Nexys 4 DDR board with Artix-7 FPGA (device xc7a100t, package csg324, speed − 1) [37]. Verilog Hardware Description Language (HDL) is used for design purposes, and Electronic design automation (EDA) Xilinx ISE 14.7 design suite [38]. For further analysis, we used the Xilinx power analysis tool and Chipscop-Pro [39].

The implementation cost of the PUF design is shown in Table 1. The design used only a fraction (8%) of the FPGA slice of the device (Artix-7 FPGA), which is negligible for large FPGAs stationed for high-end applications. Table 1, reports the size of bitstream required to reconfigurable PUF, which is relatively small. A difference based partial reconfiguration methodology is used for PUF reconfiguration over the network [40]. Additionally, new FPGA tools (Partial Reconfiguration flow in Vivado Design Suite) provided specific flow implementations for dynamic partial reconfiguration. In this work, an 8 bit configuration was followed for the Internal Configuration Access Port (ICAP) and a clock rate of 100MHz. The above settings enabled DPR to be completed in microseconds, which proved to be a real-fit for cloud-based applications.

The proposed device parameter verification circuitry is shown in Figure 2 using a Xilinx Nexys 4 DDR board, Artix-7 FPGA (device xc7a100t, package csg324, speed -1). Neural activities were monitored using an *Integrated Logic Analyzer* (ILA). The *Xilinx Power Estimation and Analysis Tools* and *Timing Closure and Design Analysis* are used additionally. We report hardware and power footprints in Table 2.

The Gaussian function representing transmission probability between LAYER-1 and LAYER-2 was implemented using linear approximations to minimize the hardware overhead. The neurons used were deployed using Leaky-Integrate and Fire(LIF) neuron models [41], as they are computationally efficient for hardware implementations. Hardware utilisation and synapses increased, which operated based on a BCM-STDP rule. Alternate synaptic rules such as Spike Driven Synaptic Plasticity (SDSP) [42]-based synaptic rule will be the focus of future investigations. These have the potential to lower synaptic weight from 32-bits to perform 1-bit operations. Assessing the practicality of the proposed system in real-world scenarios [43], the overall performance of the system was explored through a Python/SimPy [44] simulation model.

The model developed comprised of five cascaded servers as per the scheme described earlier in this paper, and each server was configured to have a log-normally distributed processing delay with a mean delay of 50 ms (standard deviation = 10 ms). The simulation was run for 1000 s of simulation time for each separate load on the system measured as the number of authentication requests submitted per second (RPS). The authentication requests were modelled as having a Poisson distribution. End-to-end delay data was collected for each authentication transaction in a run, and the following figures present histograms of the delays. Histograms were chosen as for a user of the system the mean or median delay only presents a very limited view of the actual performance that will be delivered to an individual user. The adoption of histograms to show the distribution of delays provides users with a far better indication of the range of performance that they will experience across a large number of authentication requests.

From Figure 3, Figure 4, Figure 5, Figure 6, Figure 7 and Figure 8 as presented here, some conclusions about the performance of the system can be drawn. The definition of acceptable performance in terms of authentication delay is, of course, subjective. For the sake of this discussion we will take a somewhat arbitrary position that if the overwhelming majority of requests are serviced in less than 1 s then performance is deemed acceptable.

At low loading (10 RPS), all requests are serviced within our 1 sec limit, while at a higher rates of 15 RPS and 16 RPS, progressively more requests take longer than our 1 s target, but overall performance could still be deemed acceptable.

However, at 17 RPS a very significant fraction of requests take longer than 1 s to service, with the some having to wait over 3 s. At 19 RPS, the system effectively fails, and cannot cope with the volume of traffic. This is consistent with expectations as a cascade of five stages with a fixed service time of 50 ms each per request should cope with 20 uniformly distributed RPS with an end-to-end delay of 1.25 s. The breakdown in performance in the simulated system is due to the randomness in timing of request generation and the randomness in processing time at each node.

## 6. Conclusions

This article we extends previous work on implementing multiple layers in cloud security. We add hardware security primitives and trojan detection units in combination, for a robust, multi-layer security architecture. The proposed work describes a PUF-based system with a brain-inspired device parameter analysis unit that demonstrates the ability for attack prevention from both external and internal attacks of interest primarily in the IIoT context. A vast array of surreptitious activity may be isolated to withstand a number of attack vectors, by considering security at every cloud- abstraction layers. Multiple layers of packet inspection secure the IIoT application against other opponents who may concurrently implement many tools and approaches to compromise the system security.

The security is primarily maintained by PUF-based security protocols that rely on unique device fingerprints which are hard to be compromised. The inherent flexibility and scalability provided by DPR capability with plug-and-play of new security primitives is a vital advantage of the proposed approach providing a promising direction for 5G enabled IIoT. The DPR facility removes constraints of security functions, which later is replaceable with the more secure ones.

Additionally, the hardware device parameter inspection avoids further attacks on the IIoT application using parameter variations. The continued expansion and accessibility of IIoT hardware requires flexible hardware programmability as provided by novel FPGAs. In addition, embedding analytics functions into industrial organisations requires high computational capabilities and flexible architectures provided by FPGAs. To provide satisfactory operations for the communication demand, IIoT devices with high-speed 5G networking technology is a requirement.

At the same time, appreciating the flexibility, security cannot be compromised in the infrastructure. In order to guarantee security within the model, we proposed to use primitive hardware security, for example, PUFs. The monitoring of client IIoT side-channel parameters also enhances security. In the IIoT cloud era, all software and hardware innovations have to operate together to ensure better security; failing to function might be catastrophic.

## Figures and Tables

**Figure 1 sensors-20-01963-f001:**
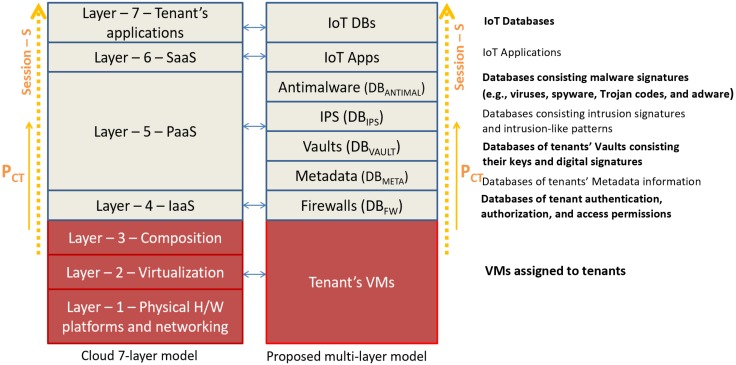
Multi-layer security proposed model [2].

**Figure 2 sensors-20-01963-f002:**
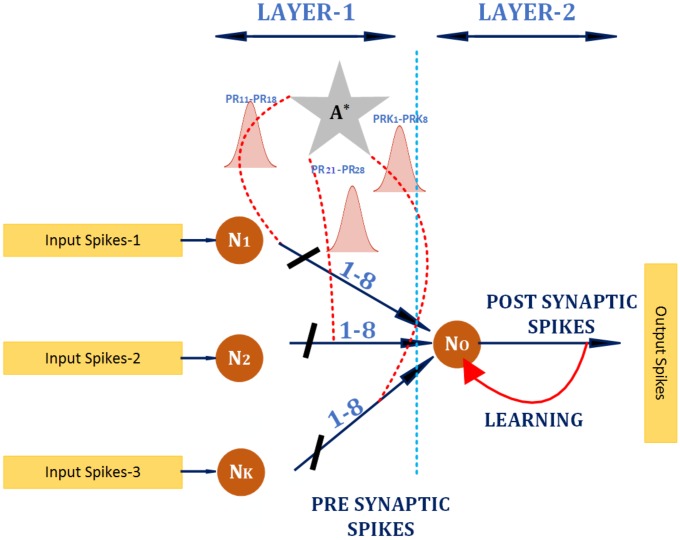
Parameter verification using spiking neural network.

**Figure 3 sensors-20-01963-f003:**
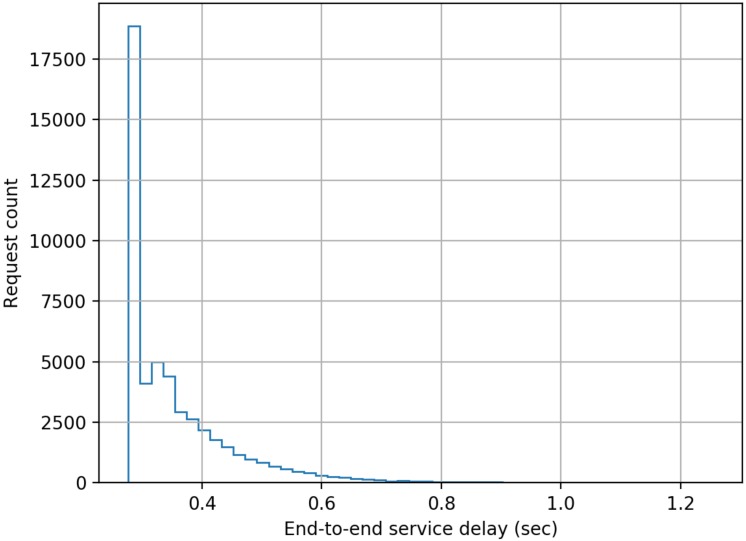
Distribution of end-to-end service delay at a mean request rate of 10 requests-per-second.

**Figure 4 sensors-20-01963-f004:**
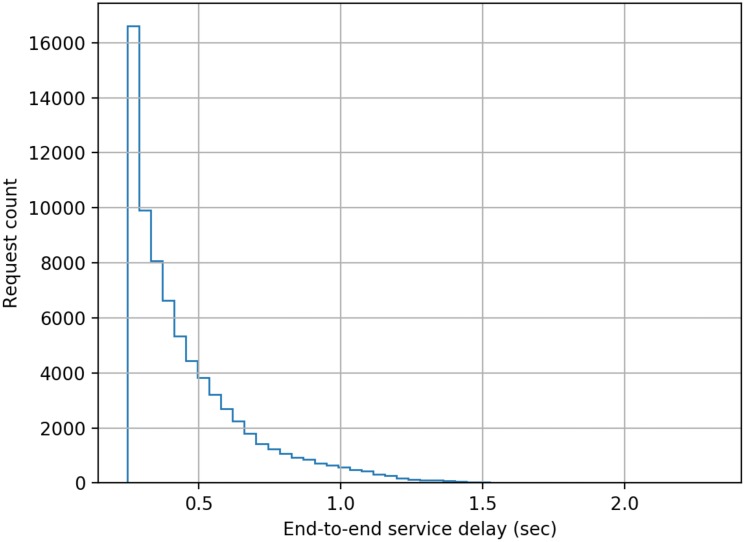
Distribution of end-to-end service delay at a mean request rate of 15 requests-per-second.

**Figure 5 sensors-20-01963-f005:**
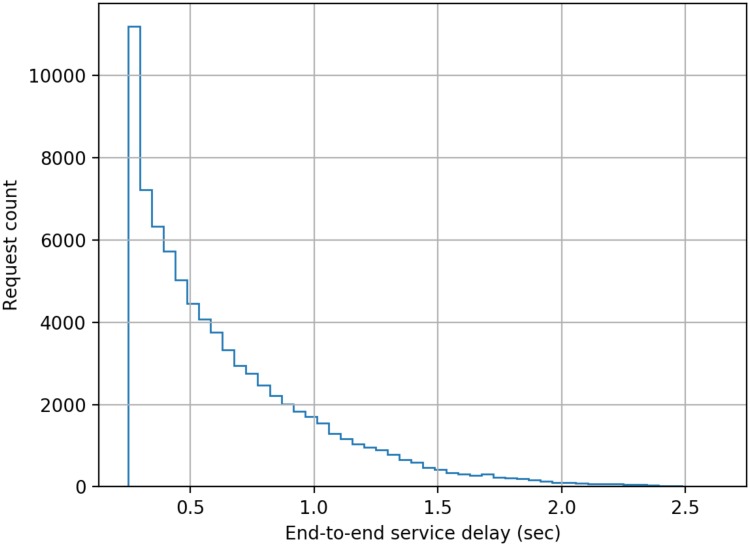
Distribution of end-to-end service delay at a mean request rate of 16 requests-per-second.

**Figure 6 sensors-20-01963-f006:**
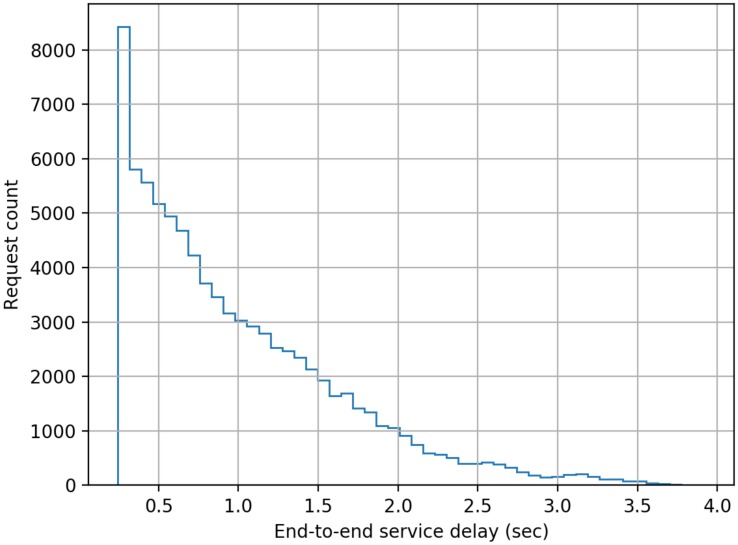
Distribution of end-to-end service delay at a mean request rate of 17 requests-per-second.

**Figure 7 sensors-20-01963-f007:**
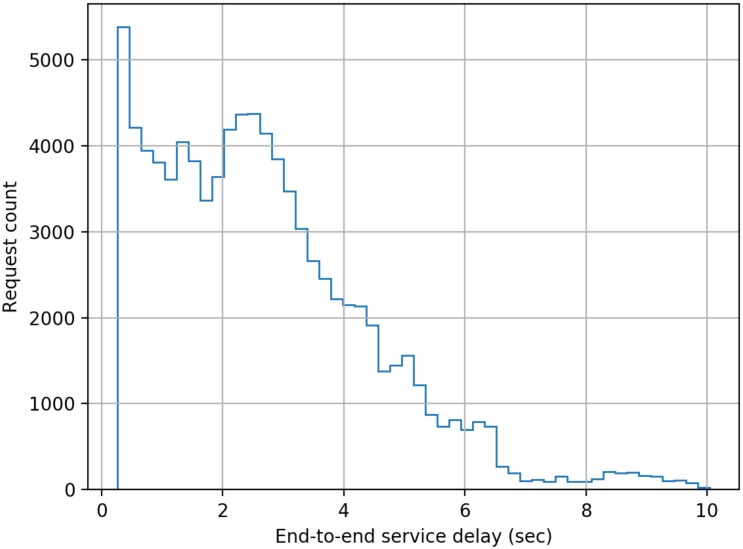
Distribution of end-to-end service delay at a mean request rate of 18 requests-per-second.

**Figure 8 sensors-20-01963-f008:**
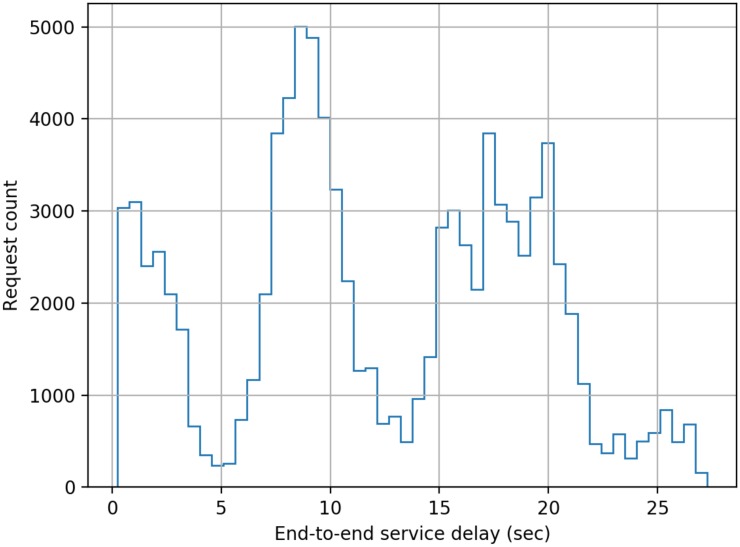
Distribution of end-to-end service delay at a mean request rate of 19 requests-per-second.

**Table 1 sensors-20-01963-t001:** Implementation Overhead [2].

Hardware Consumption *	Slice 1291	Slice Reg 10	LUTs 1282
Power Consumption			0.082 W
Bitstream Size			3737 KB

* Note The design does not contain any LUTRAMs, BRAMs/FIFOs, DSPs or buffers.

**Table 2 sensors-20-01963-t002:** Implementation Overhead of the device parameter verification unit presented in Figure 2.

Parameter/Components Hardware Consumption	Slice 14,471	Slice Reg 33,707	LUT 25,065	DSP 30
Total On-chip Power	0.082 W

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
