# Peer review of "Hardware-Intrinsic Multi-Layer Security: A New Frontier for 5G Enabled IIoT"

_sensors, 2020, doi:10.3390/s20071963_

Round 1

Reviewer 1 Report

  1. There is plagiarism of 13% from a single source, which is violating the policy.
  2. Figure 1, must be improved in terms of labeling and formatting.
  3. Algorithms are inserted as an image, which drops the paper quality.
  4. The caption of figure 2 is too long.
  5. The second paragraph is copied from a single source.
  6. Need more explanation of the NIST model and its workflow.
  7.  

Author Response

Response to Reviewer 1 Comments

Point 1: There is plagiarism of 13% from a single source, which is violating the policy.

Response 1:  We appreciate the comments. We agree with the reviewer that we have similarity from our pervious conference paper. The paper won the best award paper at the iSCI 2019 conference. We rewrite the paper again to avoid the plagiarism.

……………………………………………………………………………………………………………………………………………………….

Point 2: Figure 1, must be improved in terms of labeling and formatting.

Response 2: We followed the reviewer's advice and improved the quality of the Figure1 in terms of labelling and formatting. Please see Figure 1, page 3.

……………………………………………………………………………………………………………………………………………………….

Point 3: Algorithms are inserted as an image, which drops the paper quality.

Response 3:  We followed the reviewer's advice. The image quality allows us to maintain the template format of the journal allowing the reader to follow the methodology easily. Therefore, the image has been retained. Please see the pages 6,7, and 8.

……………………………………………………………………………………………………………………………………………………….

Point 4: The caption of figure 2 is too long.

Response 4:  We appreciate the comments. The title is edited. Please see page 9.

……………………………………………………………………………………………………………………………………………………….

Point 5: The second paragraph is copied from a single source.

Response 5:  We appreciate the comments. We have rewritten the paragraph gain.

……………………………………………………………………………………………………………………………………………………….

Point 6: Need more explanation of the NIST model and its workflow.

Response 6:  We followed the reviewer's advice and added a paragraph at the beginning of NIST model section to briefly explain the NIST model, and its workflow. The further explanation has been added into section 3 and 3.1 in the revised paper.

…………………………………………………………………………………………………………………………………………………………….

Reviewer 2 Report

Summary:
This paper proposes a multilayer security model for 5G enabled IIoT. It comprises firewall, antimalware, and access control security services. It is proposed the use the devices' hardware attributes for user authentication and authorisation to the cloud server; through using PUFs. After authorisation, the used PUFs will be replaced for new ones generated by the cloud server. It is proposed the use of Dynamic Partial Reconfiguration on FPGAs to achieve the replacement of the PUFs.

The proposal is novel and relevant as IoT security is imperative and is currently an open problem. However, I consider that some issues must be attended in order to improve and clarify the presented work.

---------------------

The paper may be improved as follows:

- Minor grammatical errors have to be addressed.

- Even if the reader knows the subject, the authors should expand the first occurrence of any the acronyms or initialism, some of them are not expanded as an example PKI, API, VM, etc.

- References to board and tools used must be included.

- The proposed security model should be validated in terms of security.

- The requirements of PUFs to be suitable for the proposed security model should be included in the article.

- What are the advantages of brain-inspired methodologies for hardware Trojan detection over bio-inspired principles?

- Figure 2 should be explained in more detail due to the fact that it is presented in a very general way and without clarification of some nomenclature used.

- Reference 24 should be revised because it does not mention anything about the Leaky-Integrate and Fire (LIF) neuron model.

- Page 11, line 300 presents a missing or typo reference that must be revised. 

- What do you propose to solve the problem in the service delay when more than 17 requests-per-second are received?

-oOo-

Author Response

Response to Reviewer 2 Comments

Point 1: Minor grammatical errors have to be addressed

Response 1:  We appreciate the comments. Proofreading has been done according to requested amendment.

……………………………………………………………………………………………………………………………………………………….

Point 2:  Even if the reader knows the subject, the authors should expand the first occurrence of any the acronyms or initialism, some of them are not expanded as an example PKI, API, VM, etc.

Response 2:  We thank the reviewer for pointing out this. We expanded the first occurrence of all the acronyms for example IPS, SaaS, PaaS, IaasS, PKI, API, VM, etc. Please see section 3, page 4.

……………………………………………………………………………………………………………………………………………………….

Point 3:  References to board and tools used must be included.

Response 3:  We appreciate the comments. Section 5 is updated by adding 3 references to the board and tool used.

……………………………………………………………………………………………………………………………………………………….

Point 4:  The proposed security model should be validated in terms of security.

Response 4:  We appreciate the comments. The validation of results is an essential part of the research process. The authors has validated the results in two ways. First, the correctness of framework has been verified theoretically (Mathematical model).

Second, we implemented an XOR PUF.  The implementation cost of the PUF design described in terms of hardware resources consumed, total on-chip power, and the configuration bitstream size, are shown in Table 1 and 2. In this work, we implemented an XOR PUF.

In order to dimension the potential practicality of the proposed system in real-world scenarios, the overall performance of the system was explored through a Python/SimPy simulation model.

……………………………………………………………………………………………………………………………………………………….

Point 5: The requirements of PUFs to be suitable for the proposed security model should be included in the article.

Response 5:  We appreciate the comments. The requirements for the PUF considered in this work include (A) A Strong PUFs with a vast number of possible challenges (B) Unpredictability of challenge responses, which means the difficulty to extrapolate or predict the CRPS from the known CRPs. The above details are included in Section 4.1.

……………………………………………………………………………………………………………………………………………………….

Point 6:  What are the advantages of brain-inspired methodologies for hardware Trojan detection over bio-inspired principles?

Response 6:  We appreciate the comments. Bio-inspired computing develops computational models using various models of biology. Brain-inspired computing is a subset of bio-inspired computing, which is mainly based on the mechanism of the brain. Brain-inspired models help to narrow the hardware trojan detection process based on the mechanism of the brain, which produces a compact computational model rather than the complex biological process involved in the former. The above details are added in Section 4.2.

……………………………………………………………………………………………………………………………………………………….

Point 7:  Figure 2 should be explained in more detail due to the fact that it is presented in a very general way and without clarification of some nomenclature used.

Response 7:  We appreciate the comments. We have updated section 4.2 to expand description of Figure 2.

……………………………………………………………………………………………………………………………………………………….

Point 8:  Reference 24 should be revised because it does not mention anything about the Leaky-Integrate and Fire (LIF) neuron model.

Response 8:  We thank the reviewer for pointing out this. The references are updated to the relevant reference

……………………………………………………………………………………………………………………………………………………….

Point 9:  Page 11, line 300 presents a missing or typo reference that must be revised. 

Response 9:  We thank the reviewer for pointing out this. The typo is updated with a relevant reference

……………………………………………………………………………………………………………………………………………………….

Point 10:  What do you propose to solve the problem in the service delay when more than 17 requests-per-second are received?

Response 10:  We appreciate the comments. One of the objectives of the simulation was to explore the performance limits of the proposed system.  If a higher request rate has to be accommodated, then either more servers, or a faster server would have to be deployed.

……………………………………………………………………………………………………………………………………………………….

Round 2

Reviewer 2 Report

Summary:
This paper proposes a multilayer security model for 5G enabled IIoT. It comprises firewall, antimalware, and access control security services. It is proposed the use the devices' hardware attributes for user authentication and authorisation to the cloud server; through using PUFs. After authorisation, the used PUFs will be replaced for new ones generated by the cloud server. It is proposed the use of Dynamic Partial Reconfiguration on FPGAs to achieve the replacement of the PUFs.

I consider that the required corrections have been made as well as the reported issues have been clarified.

.oOo.